# J-aggregates of *meso*-[2.2]paracyclophanyl-BODIPY dye for NIR-II imaging

Kang Li[1,4], Xingchen Duan[2,4], Zhiyong Jiang[1], Dan Ding [2], Yuncong Chen [3✉], Guo-Qiang Zhang [2✉] & Zhipeng Liu [1✉]

J-aggregation is an efficient strategy for the development of fluorescent imaging agents in the second near-infrared window. However, the design of the second near-infrared fluorescent J-aggregates is challenging due to the lack of suitable J-aggregation dyes. Herein, we report *meso*-[2.2]paracyclophanyl-3,5-bis-*N,N*-dimethylaminostyrl BODIPY (PCP-BDP2) as an example of BODIPY dye with J-aggregation induced the second near-infrared fluorescence. PCP-BDP2 shows an emission maximum at 1010 nm in the J-aggregation state. Mechanism studies reveal that the steric and conjugation effect of the PCP group on the BODIPY play key roles in the J-aggregation behavior and photophysical properties tuning. Notably, PCP-BDP2 J-aggregates can be utilized for lymph node imaging and fluorescence-guided surgery in the nude mouse, which demonstrates their potential clinical application. This study demonstrates BODIPY dye as an alternate J-aggregation platform for developing the second near-infrared imaging agents.

[1] College of Materials Science and Engineering, Nanjing Forestry University, Nanjing, China. [2] Key Laboratory of Bioactive Materials, Ministry of Education, and College of Life Sciences, Nankai University, Tianjin, China. [3] State Key Laboratory of Coordination Chemistry, School of Chemistry and Chemical Engineering, Nanjing University, Nanjing, China. [4]These authors contributed equally: Kang Li, Xingchen Duan. ✉email: chenyc@nju.edu.cn; guoqiangzhang@mail.nankai.edu.cn; zpliu@njfu.edu.cn

Organic fluorescent dyes with emission wavelength in the second near-infrared (NIR-II, 1000−1700 nm) window have great promise for optical imaging because of their good biological compatibility, deep tissue penetration, high-imaging resolution, and low auto-fluorescence[1–4]. Attempts to achieve NIR-II emission in organic dyes have focused mainly on molecular engineering strategy by creating large π-conjugate structure and installing strong electron-donating (D) and electron-accepting (A) groups. However, NIR-II emitting dyes are difficult to attain due to the lack of suitable π-conjugate scaffolds[4,5]. To date, NIR-II dyes are mainly derived from the polymethine skeleton[6–11] and benzobisthiadiazole core[2,12–18], and rare examples are developed based on BODIPY[19–22], squaraine[23], as well as rhodamine[24]. In this context, exploring an alternative avenue to access NIR-II dyes is highly required.

A complementary approach is to create NIR-II fluorescent J-aggregates. J-aggregates, in which the transition dipole moments of individual molecules are in slip-stacked alignment, usually display different photophysical properties from those of monomers including red-shifted absorption and fluorescence spectra and enhanced quantum yields[25–27]. Benefiting from these unique characteristics of J-aggregates, J-aggregation is becoming a facile way to achieve NIR-II fluorescence. Elegant examples of NIR-II emissive J-aggregates of cyanine and squaraine dyes were given by Chen et al.[28] and Sun et al.[23,29]. However, advances in the J-aggregates with NIR-II emission are rather slow in comparison with fluorophores developed based on molecular engineering strategy due to the lack of diversified J-aggregate fluorescent dyes. J-aggregation requires molecules to pack in a slipped arrangement, however, fluorescent dyes with large π-conjugated structure tend to stack in a face-to-face packing mode (H-aggregation)[30]. As a result, rare fluorescent dyes show J-aggregation behavior except for cyanine, perylenediimide, chlorophylls, and squaranines[23,27–29,31–35].

BODIPY (4,4-difluoro-4-bora-3a,4a-diazas-indacene) dyes are well-known fluorescent dyes with excellent photophysical properties that enable them to be good candidates for biological sensing and imaging[36,37]. Notably, recent studies revealed that BODIPY dyes show potential to be employed as J-aggregation scaffolds. Many BODIPY dyes have been reported to be J-aggregated in lipid vesicles[38–43], aqueous solution[44–48], and crystalline state[49–51]. Moreover, several examples of BODIPY J-aggregates have been used for biological sensing and imaging. For example, Kim et al.[46] developed a J-aggregating probe based on meso-ester-substituted BODIPY for tracking Eosinophil peroxidase activity in cancer cells (Supplementary Table 1). Cheng et al.[42] prepared aza-BODIPY J-aggregate-containing liposomal nanoparticles (NPs), which could be used for NIR-I cancer imaging. Su et al.[43] reported nano-assemblies of J-aggregated BODIPY as a stimuli-responsive tool for imaging oxidative stress in living systems. Although numbers of fluorescent BODIPY J-aggregates have been reported, understanding the relationship between the structure and J-aggregation, as well as overcoming the aggregation-caused emission (ACQ) quenching, is still required. Furthermore, exploring building blocks for NIR-II emissive BODIPY J-aggregates remains a great challenge. In the course of our continuous efforts in the development of solid-state and NIR-II emissive BODIPY dyes[21,52,53], we envisioned that J-aggregation of BODIPY dyes could be an efficient strategy to achieve NIR-II emission. Herein, we report a BODIPY dye (PCP-BDP2) that readily J-aggregates in an aqueous solution by introducing [2,2]paracyclophane (PCP) group to the meso-position of BODIPY (Fig. 1). PCP-BDP2 J-aggregates show both NIR-I ($J_1$-band, 900 nm) and NIR-II ($J_2$-band, 1010 nm) emission in the aggregation state, and only NIR-II emission in the crystalline powder state. By co-precipitation PCP-BDP2 with Pluronic F-127, the J-aggregates are stabilized in the assembled NPs and

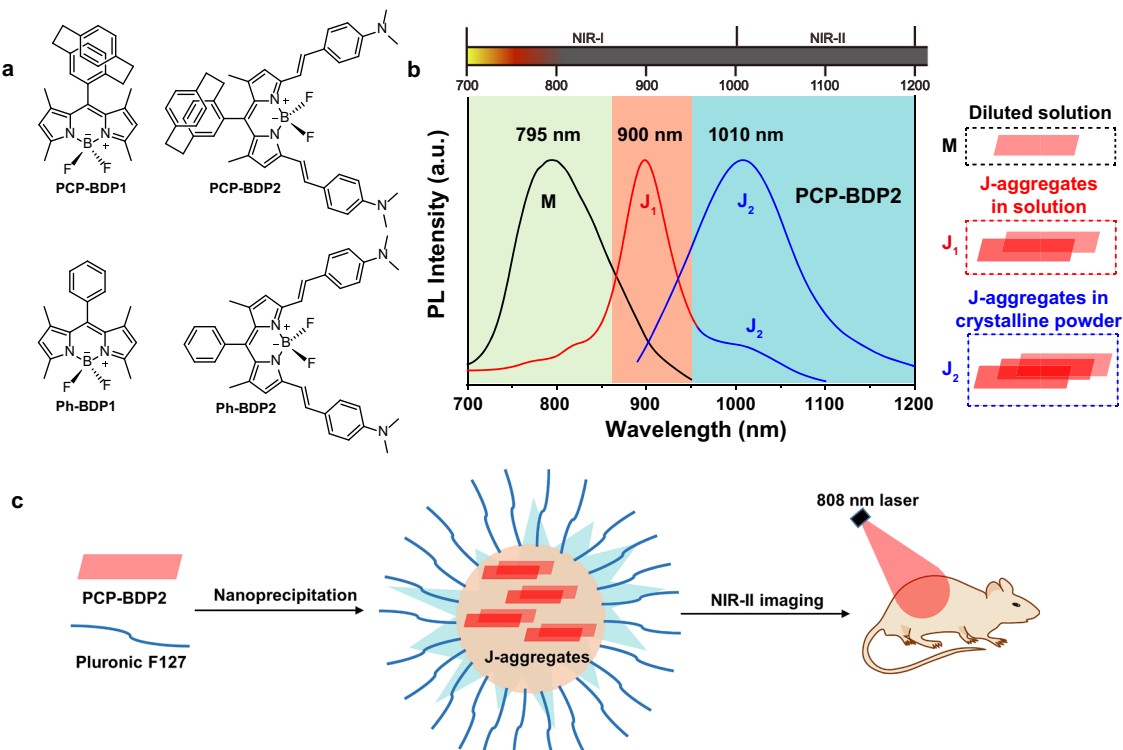

**Fig. 1 Molecular structures and working principle. a** Chemical structures of PCP-BDP1, PCP-BDP2, Ph-BDP1, and Ph-BDP2. **b** Fluorescence spectra of PCP-BDP2 in diluted dichloromethane solution (10 μM) (black line, "M" refers to monomer), THF-water binary solvents (10 μM, 1:9, v/v) (red line, $J_1$-band), and the crystalline powder (blue line, $J_2$-band). The parallelogram in red represents the single molecular of PCP-BDP2. **c** Schematic illustration of the construction of the PCP-BDP2 nanoparticles for NIR-II imaging. Source data underlying (**b**) are provided as a Source data file.

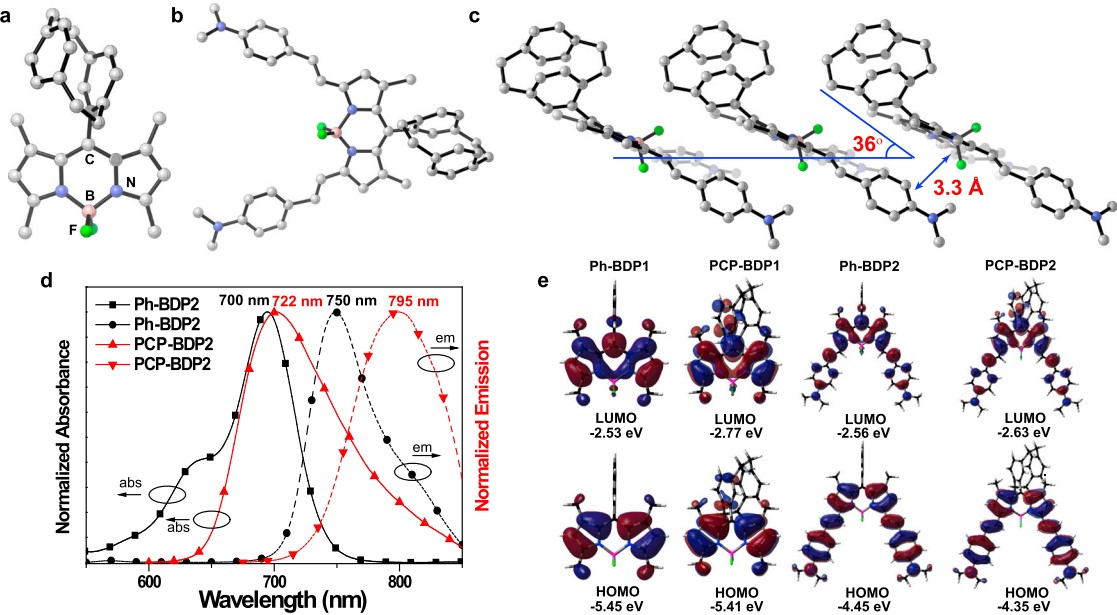

**Fig. 2 Crystal structures and photophysical properties.** Ball and stick illustration of the X-ray structure of PCP-BDP1 (**a**) and PCP-BDP2 (**b**). **c** Molecular packing diagram of PCP-BDP2. Solvent molecules and H atoms are omitted for clarity. **d** Normalized absorption and emission spectra of Ph-BDP2 (black line) and PCP-BDP2 (red line) in DCM (10 μM). "abs" and "em" refer to "absorption" and "emission," respectively. **e** Calculated frontier molecular orbitals for Ph-BDP1, PCP-BDP1, Ph-BDP2, and PCP-BDP2 and their orbital energies in the optimized excited state. Source data underlying (**d**) are provided as a Source data file.

show bright NIR-II emission with high fluorescence quantum yield ($\Phi_f$) of 6.4%. We also demonstrated the promising NIR-II imaging ability of the J-aggregates both in vitro and in vivo, as well as the potential application in the clinic for fluorescence-guided surgery (FGS) in nude mice.

## Results

**Design and synthesis.** Due to the large π-conjugated frameworks, BODIPY dyes tend to aggregate in a face-to-face stacking mode (H-aggregation), and J-aggregation is usually not favored. As a result, most BODIPY dyes usually display ACQ in the aggregation state. To dismiss the strongly π–π interactions between the indacene plane and subsequently achieve the goal of J-aggregation, we introduced a PCP group with a three-dimension structure, to the *meso*-position of the BODIPY core. Moreover, we conjugated the strong electron-donating group, *N,N*-dimethylaminostyryl, to the 3,5-position of the PCP-BODIPY structure to realize the NIR-I emission (Fig. 1). Take advantage of red-shifted emission, J-aggregates of PCP-BOIDPY dye are expected to be fluorescent in the NIR-II region. The synthesis of PCP-BDP1 and PCP-BDP2 is outlined in Supplementary Fig. 1. Additionally, two BODIPY dyes, Ph-BDP1[51] and Ph-BDP2[54], which have the phenyl group on the *meso*-position of the BODIPY core, were synthesized for comparison.

**X-ray single-crystal structure analysis.** To confirm the molecular design concept, we first investigated the molecular packing mode of PCP-BDP1 and PCP-BDP2 via single-crystal structure analysis (Supplementary Table 2). In the single-crystal structure, the indacene planes of PCP-BDP1 and PCP-BDP2 are slightly bent due to the weak intermolecular interactions (Fig. 2 and Supplementary Fig. 2). The PCP group is highly twisted to the BODIPY core. The dihedral angles between the phenyl ring that was attached to the BODIPY core and the indacene plane are ranging from 55.5° to 58.5°. In the molecular packing structure of PCP-BDP1, PCP groups are connected to the indacene plane through

C–H…π (~3.8 Å) interactions. Due to the steric effect of the PCP group, the face-to-face molecular packing mode between the indacene planes is disfavored, as a result, PCP-BDP1 molecules are J-aggregated with a slipping angle of 38°[27]. The distances between the borondipyrrole planes are ~3.7 Å. Ph-BDP1 show different molecular packing mode comparing to PCP-BDP1[51]. As shown in Supplementary Fig. 3, Ph-BDP1 molecules are arranged in a zigzag pattern through weak C–H…π interactions and no J-dimers were observed in the molecular packing structure. This difference demonstrates that the PCP group plays a key role in tuning the J-aggregation of the BODIPY core. In PCP-BDP2, π–π interactions between the *N,N*-dimethylaminophenyl group and the indacene plane, the C–H…F hydrogen bond (~3.4 Å) between the PCP group and the borondipyrrole plane, and C–H…π (~3.8 Å) interactions between the PCP group and the *N,N*-dimethylaminophenyl group dominate the molecular packing structure of PCP-BDP2, which facilitate the J-aggregation packing mode. The slipping angle and the distance between each molecule are determined to be 36° and ~3.3 Å, respectively (Fig. 2c).

**Photophysical properties.** Before investigating the photophysical properties of J-aggregates, first, we measured the absorption and emission spectra of PCP-BDP1 and PCP-BDP2 in diluted solutions. In dichloromethane (DCM), PCP-BDP1 displayed the main absorption band ($\lambda_{abs}$) centered at 523 nm ($\varepsilon = 41,900$ M$^{-1}$ cm$^{-1}$), and two emission bands centered at 524 and 554 nm, respectively (Supplementary Fig. 4). The sharp and narrow emission band at 524 nm can be assigned as the typical local excite (LE) emission of the BODIPY core, while the broad and weak emission band at 554 nm should be ascribed to the charge transfer (CT) emission band. In most cases, the phenyl ring at the *meso*-position has a very weak conjugation effect on the BODIPY core because of its perpendicular orientation toward the BODIPY core[55]. For example, Ph-BDP1 only shows the LE emission at 515 nm in DCM, and the CT band around 550 nm is hardly to be distinguished from the LE emission. However, the free rotation of the PCP group in PCP-BDP1 is inhibited due to the steric effect of the PCP group, leading to the CT

from the PCP group to the BODIPY core. It is worth noting that the CT emission intensity of PCP-BDP1 is highly dependent on the viscosity of the solvent. As shown in Supplementary Fig. 5, the emission intensity at 554 nm is gradually increased with the viscosity increasing from 0.6 to 630 cP, while the intensity at 524 nm is almost unchanged. This result suggests that the CT process was greatly enhanced due to the inhibited intramolecular motion of PCP-BDP1 in the high viscosity media. On the other hand, PCP-BDP2 showed the absorption and emission bands centered at 722 and 795 nm, respectively, which is red-shifted in comparison with that of Ph-BDP2 ($\lambda_{abs} = 700$ nm and $\lambda_{em} = 750$ nm in DCM, Fig. 2d). This result suggests that the CT process is more strengthened in PCP-BDP2 than in Ph-BDP2, which also confirms the electron-conjugating effect of the PCP group on the BODIPY core.

Both the absorption and emission wavelength of PCP-BDP1 remain almost unchanged in different solvents, and only a slight increase of the absorbance and the emission intensity is observed in a polar solvent such as methanol and dimethylsulfoxide (DMSO) (Supplementary Fig. 6). All of these results demonstrate the weak CT of PCP-BDP1. Due to the donor (D)−π−acceptor (A) structure, PCP-BDP2 shows polarity dependent absorption and emission (Supplementary Fig. 7). The absorption and emission bands are red-shifted with the increased solvent polarity. For example, PCP-BDP2 exhibited $\lambda_{em}$ in DMSO at 824 nm, which is 73 nm red-shifted relative to the $\lambda_{em}$ of 751 nm in hexane. Interestingly, the emission intensity increment is also observed in solvents with relatively large viscosity, such as o-dichlorobenzene (1.32 cP at 25 °C), 1,4-dioxane (1.20 cP at 25 °C), and DMSO (1.98 cP at 25 °C). This result suggests that the non-radiative decay process induced by intermolecular motions is inhibited in these solvents with higher viscosity. Fluorescence changes in methanol with a varied volume of glycerol further demonstrated the solvent- and viscosity-dependent fluorescence of PCP-BDP2 (Supplementary Fig. 8). With the viscosity increasing from 0.6 to 130 cP, PCP-BDP2 showed a 1.4-fold fluorescence intensity increase accompanied by the red-shift of $\lambda_{em}$ from 790 to 810 nm. The increase in viscosity causes further red-shift of $\lambda_{em}$ and quenched fluorescence intensity. The decreased solubility in glycerol may responsible for the emission quenching.

We next carried out the theoretical calculation to understand the different absorption and emission properties between these PCP and phenyl groups substituted BODIPY dyes (Supplementary Table 3). Time-dependent density functional theory results show that the main absorption and emission bands of Ph-BDP1 are contributed mainly by the transition from the highest occupied molecular orbital (HOMO) to the lowest unoccupied molecular orbital (LUMO) (Fig. 2e and Supplementary Fig. 9a). Moreover, the HOMO and LUMO are delocalized on the indacene plane, while the phenyl ring at the meso-position has negligible contribution to the electronic structure of Ph-BDP1 in both ground and excited states. Different from that, the PCP group in PCP-BDP1 is conducive to the electron delocalization in both HOMO and LUMO (Fig. 2e and Supplementary Fig. 9b). This difference makes the main absorption band of PCP-BDP1 that is composed of both the first ($S_1$) and second ($S_2$) excited states, which mainly originated from the HOMO → LUMO, HOMO-1 → LUMO, and HOMO-3 → LUMO transitions. The electronic transition of $S_1 \rightarrow S_0$ (ground state, $f = 0.0900$) can be ascribed to the CT emission. This result is consistent with the fluorescent spectrum of PCP-BDP1 obtained in DCM. The CT process from the PCP group to the indacene plane should be responsible for the different emission behavior of Ph-BDP1 and PCP-BDP1. Interestingly, the main absorption and emission bands of both Ph-BDP2 and PCP-BDP2 are contributed by the HOMO → LUMO transitions (Fig. 2e and Supplementary

Fig. 10). The HOMO and LUMO of Ph-BDP1 and the HOMO of PCP-BDP2 are delocalized on the whole molecular skeleton except for the phenyl and PCP groups. However, the LUMO of PCP-BDP2 is localized on the BODIPY core as well as the meso-phenyl ring of the PCP group. This feature favors the strengthened CT process, and therefore, the red-shifted absorption and emission bands of PCP-BDP2 in comparison with that of PCP-BDP1 can be rationalized.

Furthermore, we analyzed the optimized geometries in both ground and excited states to explore whether the photophysical properties difference is the result of geometry rearrangement in the excited state (Supplementary Fig. 11). Like most classical BOPIPY dyes, Ph-BDP1 and Ph-BDP2 show almost the same geometries in both ground and excited states: the whole molecule including the indacene plane and the 4-N,N-dimethylstyl group are well planar, while the phenyl group at the meso-position is perpendicular to the indacene plane. In contrast, an excited state geometry rearrangement was observed for PCP-BDP1 and PCP-BDP2. In the ground state, the indacene plane of PCP-BDP1 and PCP-BDP2 is significant bending to accommodate the van der Waals repulsion of the PCP group and the methyl groups at 1,7-positions. As a result, the boron atom as well as the carbon atom at PCP group ($C_{PCP}$), which attached to the meso-position of the BODIPY core, deviated from the indacene plane with a distance around 0.26 and 0.49 Å for PCP-BDP1, and 0.62 and 0.44 Å for PCP-BDP2, respectively. In the excited state, the indacene plane is further bent. The deviation of the boron and $C_{PCP}$ atoms increased to 0.38 and 0.84 Å for PCP-BDP1, and 0.79 and 0.45 Å for PCP-BDP2, respectively. This rearrangement would ultimately favor the thermal relaxation of PCP-BDP1 and PCP-BDP2 in the first excited singlet state to an energetically hot, ground-state species. When PCP-BDP1 and PCP-BDP2 are in a highly viscous media, such geometry rearrangement can be efficiently inhibited. As a result, fluorescence intensity increment can be observed.

**J-aggregation behavior in the aggregated state**. To further explore the J-aggregation behavior of PCP-BDP1 and PCP-BDP2, we investigated their emission variation in the tetrahydrofuran (THF)-water binary solvents. In THF, PCP-BDP1 and PCP-BDP2 show one sharp absorption band at 523 and 718 nm, respectively. After the addition of water to the solution, the absorbance decreases and red-shifted to 532 and 748 nm when the water volumetric factions ($f_w$) increased to 99% (Supplementary Fig. 12). Moreover, this red-shift was also observed in the absorption spectra in the film state (Supplementary Fig. 13). All these results indicate the formation of J-aggregates. In the emission spectra, the emission intensity at 520 and 560 nm of PCP-BDP1 remain unchanged with $f_w$ increasing from 0 to 70%. With $f_w$ of 80%, the emission intensity at 520 nm was distinctly increased, accompanied by the slightly red-shifted CT band. This result suggests that PCP-BDP1 begins to aggregate, the restricted molecular motion and enhanced CT process should be responsible for the enhanced LE emission and the red-shifted CT emission. Notably, when $f_w$ reaches 90%, the LE emission is distinctly decreased, and the CT emission is almost disappeared. A new broad emission band ranging from 560 to 850 nm, which should be assigned as the emission of J-aggregates, appears simultaneously (Fig. 3a). Since the emission wavelength of J-aggregates is highly dependent on the size of the aggregates, the coexistence of the J-aggregates with different sizes is the reason for the formation of the broad emission band[56].

PCP-BDP2 showed the same aggregation trend as PCP-BDP1. As shown in Fig. 3b, the emission intensity at 790 nm was enhanced and red-shifted to 820 nm with $f_w$ increasing from 10 to 60%. The

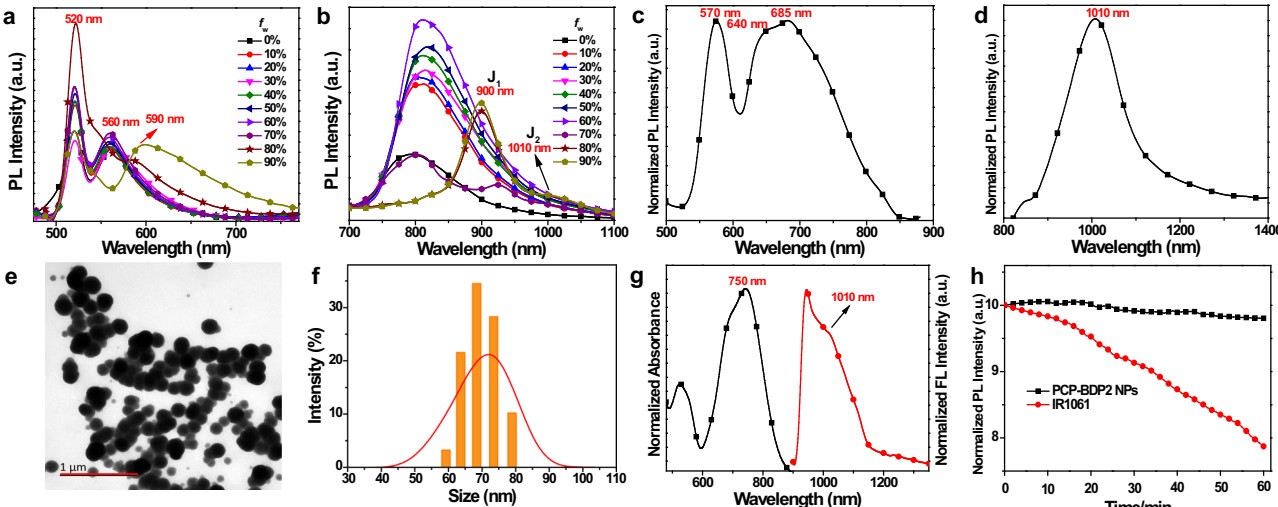

**Fig. 3 Characterization and fluorescent properties of J-aggregates.** Fluorescence spectra of PCP-BDP1 (**a**) (10 μM) and PCP-BDP2 (**b**) (10 μM) in THF-water binary solvents varied $f_w$. Fluorescent spectra of PCP-BDP1 ((**c**), excited at 480 nm) and PCP-BDP2 ((**d**), excited with 808 nm laser) in the crystalline powder state. **e** TEM images of PCP-BDP2 NPs. The results are representative of three independent experiments. **f** The particle size distribution of PCP-BDP2 NPs measured by DLS with a PDI of 0.081. **g** Normalized absorption and emission spectra of PCP-BDP2 NPs in PBS solution. **h** Normalized emission decay of PCP-BDP2 NPs and IR1061 at their emission maxima under laser irradiation (808 nm, 100 mW/cm², 60 min) for different irradiation times. Source data underlying (**a**–**d**, **f**–**h**) are provided as a Source data file.

intramolecular motion restriction and the CT process enhancement induced by the viscosity and polarity increment should be responsible for the enhanced and red-shifted emission, respectively. With $f_w$ of 70%, the emission at 820 nm was distinctly quenched, accompanied by the appearance of a new broad peak around 930 nm, suggesting the formation of J-aggregates. When $f_w$ reaches 80 and 90%, most of the PCP-BDP2 molecules may be aggregated to J-aggregate, as a result, the emission around 800 nm almost disappears, while the emission band of J-aggregates at ~900 nm ($J_1$-band) is enhanced. Notably, a weak NIR-II emission band around 1000 nm ($J_2$-band) was also observed as the side peak of the main emission band, which suggests that the emission of PCP-BDP2 J-aggregates is capable of red-shifting to the NIR-II region when the more condensed J-aggregates are formed.

To verify this theory, we further measured the emission spectra of PCP-BDP1 and PCP-BDP2 in the solid state. The crystalline powder of PCP-BDP1 showed multiple J-aggregates emission bands at 570, 640, and 685 nm, respectively, which is contributed by J-aggregates with different sizes (Fig. 3c). Notably, PCP-BDP2 displayed one broad J-aggregate emission centered at 1010 nm, which is consistent with the $J_2$-band observed in THF-water binary solvent with $f_w$ of 99% (Fig. 3d). This result confirms that the condensed molecular packing mode should be responsible for the generation of the $J_2$-band of PCP-BDP2. Additionally, we also measured the absorption and emission spectra of Ph-BDP1 and Ph-BDP2 in THF-water. With the $f_w$ increasing from 0 to 90%, the absorption bands of both Ph-BDP1 and Ph-BDP2 were red-shifted and broadened, indicating the formation of H- and J-aggregates at the same time. Different from the aggregation-caused J-aggregates emission enhancement of PCP-BDP1 and PCP-BDP2, the emission of both two compounds was gradually decreased, and no J-aggregates emission was observed (Supplementary Fig. 14). These results suggest that H-aggregation-induced emission quenching dominated the photophysical properties of Ph-BDP1 and Ph-BDP2 in the aggregation state, which further demonstrated the key role that PCP group plays in the J-aggregation behavior of PCP-BDP1 and PCP-BDP2.

Encouraged by the NIR-II emission capability of PCP-BDP2 J-aggregates observed in both THF-water and solid state, we investigated that whether the NIR-II emissive J-aggregates could

be stabilized in NPs and be employed for the NIR-II imaging. We prepared PCP-BDP2 NPs by encapsulating the PCP-BDP2 aggregates into a Pluronic F-127 matrix. The PCP-BDP2 NPs showed a spherical morphology with a diameter of ~70 nm, which was characterized using transmission electron microscopy (TEM) (Fig. 3e). The average diameter of PCP-BDP2 NPs was measured to be ~75 nm by a dynamic light scattering (DLS) experiment with a low polydispersity index (PDI) of 0.081 (Fig. 3f). In PBS buffer, the PCP-BDP2 NPs showed $\lambda_{abs}$ and $\lambda_{em}$ around 750 and 1010 nm, respectively, which is comparable with those observed in the aggregation state. This result suggests that the J-aggregates can be efficiently stabilized in the Pluronic F-127 matrix.

The PCP-BDP2 NPs showed relatively high $\Phi_f$ in PBS buffer. The calculated $\Phi_f$ value for PCP-BDP2 NPs was 6.4% (reference dye IR 26, $\Phi_f = 0.1\%$), which is higher than most reported NIR-II emissive BODIPY dyes and J-aggregates such as NJ1060 (1%)[21], and NIR-II-WAZABY-01 (0.8%)[20]. Furthermore, after continuous laser irradiation (808 nm, 100 mW/cm²) for 60 min, the NIR-II emission intensities of PCP-BDP2 NPs remained almost unchanged, however, another famous NIR-II fluorophore, IR1061, showed rapidly decayed emission from 100 to 75% under the same conditions (Fig. 3h). Moreover, PCP-BDP2 NPs showed no apparent absorption and emission spectral change in PBS in the presence of glutathione, cysteine, and hydrogen peroxide (Supplementary Fig. 15). These results demonstrate the good photo- and chemical-stability of PCP-BDP2 NPs.

**In vitro imaging.** To verify the biological imaging capability of PCP-BDP2 NPs, we carried out both in vitro and in vivo NIR-II fluorescence imaging experiments. First, we evaluated the cytotoxicity of PCP-BDP2 in L02 and HepG2 cells by standard MTT assay. As shown in Supplementary Fig. 16, PCP-BDP2 with varied concentrations from 0.4 to 51.2 μM show negligible influence on the survival of L02 and HepG2 cells, suggesting its good biosafety. In the imaging experiments, a 808 laser was used as the excitation resource because of its general availability and reduced biological absorption. A clinically approved NIR-I dye, ICG, was employed as a control. The PL intensity of PCP-BDP2 NPs with different

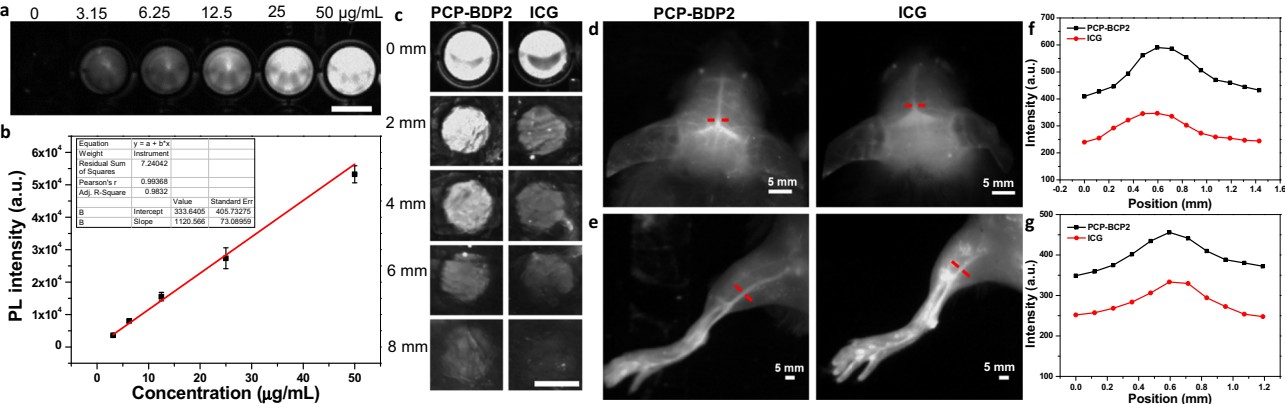

**Fig. 4 In vitro NIR-II fluorescence imaging of PCP-BDP2 NPs. a** NIR-II fluorescence images of different concentrations. Scale bar, 5 mm. **b** The quantitative relationship between NIR-II fluorescence intensity with concentrations. Bars represent mean ± SD derived from $n = 3$ independent experiments. **c** Comparison of penetration depth of PCP-BDP2 NPs and ICG (1 mg/mL, 1100 nm LP filter, 500 ms) under the different thickness of chicken tissue. Scale bar, 5 mm. NIR-II fluorescence imaging of blood vessels in **d** brain and **e** hindlimb of PCP-BDP2 NPs and ICG, respectively. **f, g** The quantification of fluorescent signals at the cross-section in hindlimb and brain (along the red-dashed line), respectively. The results are representative of three independent experiments. Source data underlying (**b**, **f**, **g**) are provided as a Source data file.

concentrations was detected, and the NIR-II emission signals of PCP-BDP2 NPs showed a linear increase with the concentration, which is consistent with the above results that NIR-II emissive J-aggregates are highly related to the degree of aggregation (Fig. 4a, b). Besides, the NIR-II fluorescence signals of PCP-BDP2 NPs and ICG under the different thickness of chicken tissues were collected, the NPs showed good NIR-II imaging penetration depth up to 8 mm, which is higher than the 6 mm of ICG (Fig. 4c). Altogether, these results indicated that PCP-BDP2 NPs hold the promise to be a superb NIR-II fluorescent dye than ICG.

**In vivo imaging.** We further carried out in vivo NIR-II imaging of cerebral vasculature and hindlimb with the PCP-BDP2 NPs. The NPs in PBS were injected into mice via the tail vein. After the injection for 5 min, the blood vessels in both the brain and hindlimb can be visualized from the surrounding background tissue with high resolution. The brightness and clarity of PCP-BDP2 NPs were higher than that of ICG with the same imaging conditions, implying that the PCP-BDP2 NPs can offer better imaging quality than ICG (Fig. 4d, f), which was quantitatively analyzed and compared (Fig. 4e, g). Moreover, the fluorescence signal of PCP-BDP2 NPs decreased gradually with the time increased from 5 min to 24 h, while the fluorescence of ICG is almost undetectable when the time increased to 8 h (Supplementary Figs. 17 and 18). Besides, the PCP-BDP2 NPs were found to accumulate in the liver and spleen, properly due to the uptake by mononuclear phagocytic system-related organs (Supplementary Fig. 19). All the above imaging results demonstrate the promising high-resolution and long-term NIR-II imaging ability of the PCP-BDP2 NPs.

**Lymph node imaging.** The lymphatic system is important for maintaining fluid homeostasis and immunity, which is increasingly considered as a conduit for the metastasis of a variety of cancers such as breast cancer, melanoma, and so on[57]. Optical imaging of the lymphatic system can map lymphatic drainage, locate sentinel lymph node (SLN), and visualize multiple lymph nodes[58,59]. In this experiment, PCP-BDP2 NPs were used for in vivo imaging to assess its ability for mapping lymph nodes. The PCP-BDP2 NPs (50 µL, 1 mg/mL) were subcutaneously injected into the footpads of nude mice, the lymphatic vasculature and the SLN were observed immediately (Fig. 5a), and it

was still clear 5 h after injection (Fig. 5b). Under the guidance of the fluorescence signal of PCP-BDP2 NPs, the SLN with a diameter of <1 mm was then removed precisely (Fig. 5c), which was further proved by the (hematoxylin and eosin (H&E)) histological staining (Fig. 5d).

**Image-guided surgery.** Recently, NIR-II fluorescence-guided cancer surgery has been proven feasible clinically, which reduce cancer recurrence and promote the outcomes of cancer surgery[4,60]. To demonstrate that the strong NIR-II fluorescence signal of PCP-BDP2 endows its ability for image-guided cancer surgery, the peritoneal carcinomatosis-bearing mouse model was established, which scatter numerous tumor nodules of various sizes in the peritoneal cavity, especially those with diameters < 1 mm. After the PCP-BDP2 NPs were intravenously injected into the mice for 24 h, the surgery was first performed by a surgeon by opening the mouse abdomen. As we selected luciferase-expressed 4T1 tumors that exhibited bioluminescence after injection with D-luciferin, so the signals of fluorescence of PCP-BDP2 NPs and the bioluminescence of luciferase were well colocalized (Fig. 6a). And then lots of large tumor nodules with diameters > 1 mm were removed by the surgeon's naked eyes. Then with the guidance of high brightness of PCP-BDP2 NPs, smaller tumor nodules were resected (Fig. 6c). The well-overlapped bioluminescence and fluorescence signals of the removed nodules indicate those were indeed tumors (Fig. 6b), which was also proved by the H&E staining (Fig. 6d), and these results together demonstrate the accuracy of operation.

**Discussion**

In summary, we have demonstrated a BODIPY dye (PCP-BDP2) with J-aggregation-induced emission in the NIR-II biological window by introducing a PCP group to the *meso*-position of 3,5-bis-*N,N*-dimethylaminostyryl BODIPY. Single-crystal X-ray structure analysis and DFT calculation reveal that the PCP group plays a key role in the photophysical property and J-aggregation tuning. Due to the conjugation effect of the PCP group, PCP-BDP2 shows $\lambda_{em}$ approximately at 795 nm, which is red-shifted comparing to its analog Ph-BDP2 ($\lambda_{em} = 750$ nm) with phenyl ring substituted at *meso*-position. PCP-BDP2 is preferred to J-aggregate in the aggregation state. Notably, PCP-BDP2 J-aggregates show both NIR-I ($J_1$-band, 900 nm) and NIR-II ($J_2$-band, 1010 nm) emission in THF-water binary solvent, and

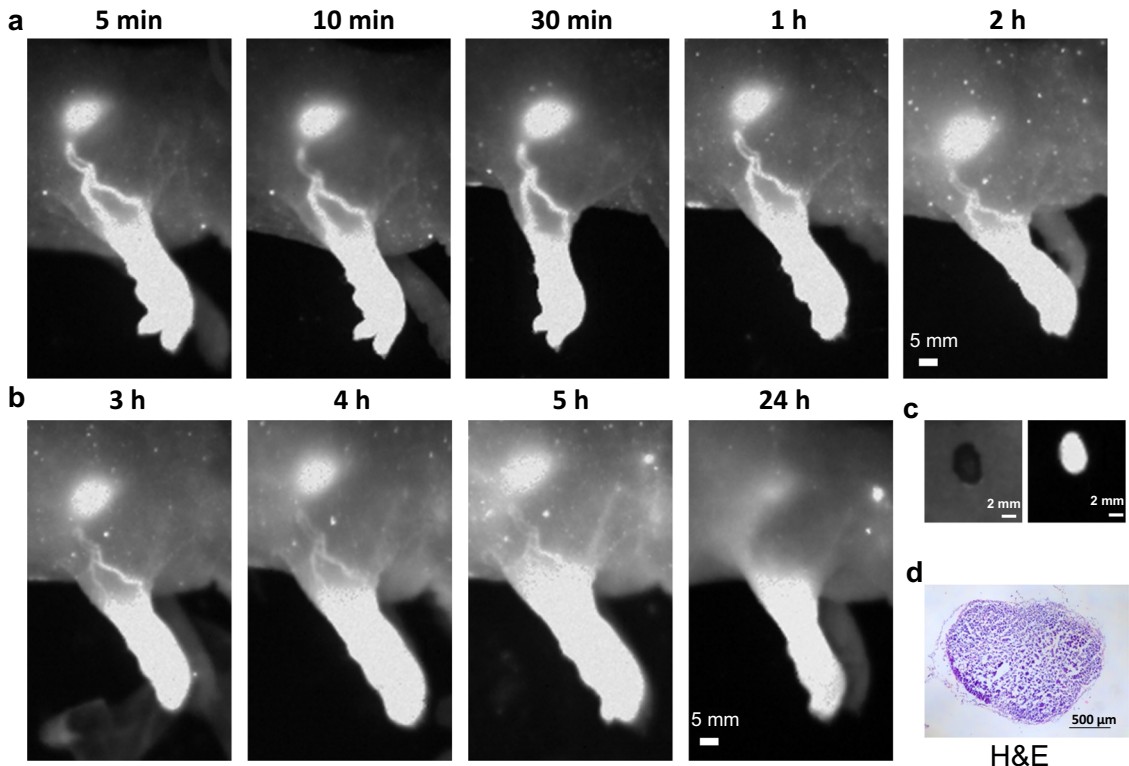

**Fig. 5 In vivo NIR-II fluorescence imaging of the lymphatic system with PCP-BDP2 NPs. a**, **b** Fluorescence images of lymph node and lymphatic vessels were captured at 5, 10, 30 min, 1–5, and 24 h, respectively. **c** Bright (left) and fluorescent (right) images of sentinel lymph node extracted from the mouse. **d** H&E staining of the excised lymph node. The results are representative of three independent experiments.

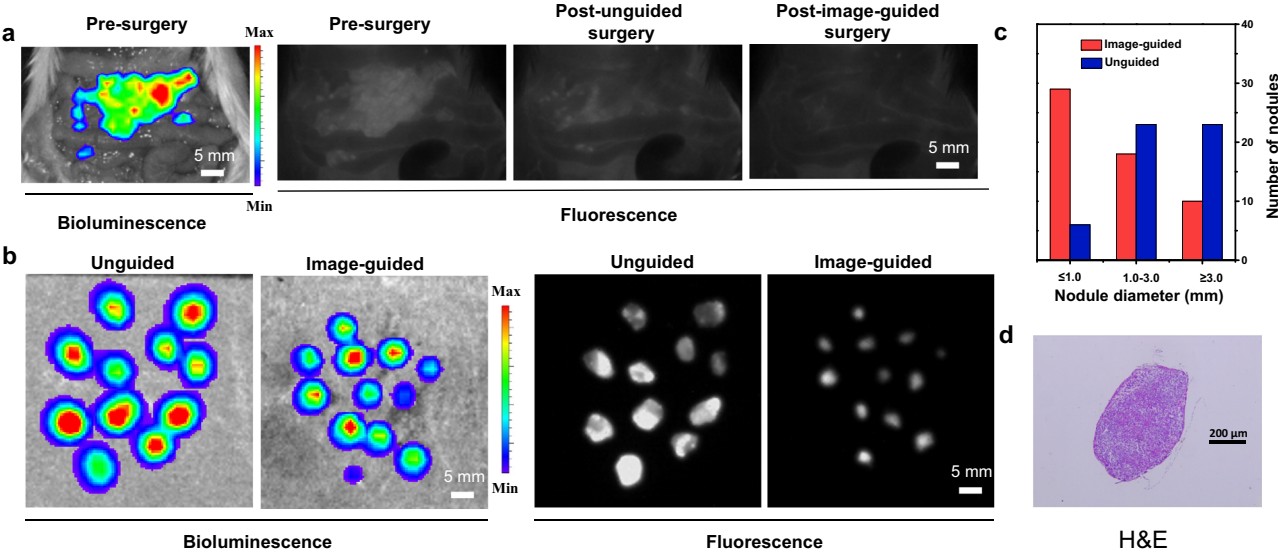

**Fig. 6 Image-guided surgery of PCP-BDP2 NPs. a** Bioluminescence and fluorescence imaging of the abdominal cavity before (presurgery) and after (post-surgery) tumor resection. **b** Bioluminescence and NIR-II fluorescence signals of the resected nodules of unguided and PCP-BDP2 NPs-guided groups. **c** Statistical chart of nodules diameters resected from fluorescence-guided and unguided groups. **d** H&E staining of excised tumor node. All the metastases were confirmed to be malignant and repeated for three times in independent experiments. Source data underlying (**c**) are provided as a Source data file.

only the NIR-II emission in the crystalline powder state. The J-aggregate of PCP-BDP2 can be efficiently stabilized in the assembled NPs, and show bright NIR-II emission with a high $\Phi_f$ of 6.4%. We also demonstrate the promising NIR-II imaging ability of the J-aggregates both in vitro and in vivo. PCP-BDP2 NPs show good photo-/chemo-stability and NIR-II imaging penetration, which facilitates its high-resolution and long-term

NIR-II imaging ability. Furthermore, we demonstrate the potential application in the clinic for lymph node imaging and FGS in nude mice. The results from this fundamental research provide insights for not only molecular engineering J-aggregation of BODIPY dyes for NIR-II imaging but also manipulating desired optical properties of luminescent dyes for potential applications in biological sensing and imaging.

## Methods

**Preparation of PCP-BDP2 NPs.** PCP-BDP2 (2 mg) was dissolved in THF (1 mL) by bath sonication. 10 mg Pluronic F-127 was dissolved in 1 mL of PCP-BDP2 stock solution, and then the solutions were added to 10 mL deionized water and dispersed by vibration for 30 min under ultrasound. After dispersion, THF was removed under reduced pressure. Then, the solutions were filtered through a filter. To remove the excess Pluronic F-127, the PCP-BDP2 NPs water dispersion was treated by the repeated centrifugal washing process thrice.

**Photostability.** The photostability of PCP-BDP2 NPs and IR1061 was investigated in PBS solution containing 5% DMSO as co-solvent with a concentration of 10 μM. The respective dye's solution in 1 cm quartz cuvettes (200 μL) was illuminated with a continuous laser (808 nm, 100 mW/cm$^2$) for 60 min. The emission intensity of PCP-BDP2 NPs and IR1061 was measured every 1 min.

**In vitro and in vivo NIR-II fluorescence imaging.** All images of fluorescence were collected on a home-built imaging set-up consisting of a 2D InGaAs camera (Princeton Instruments, 2D OMA-V). An 808 laser was used as the excitation resource, and all signals were detected utilizing an 1100 nm LP filter. The NIR-II fluorescence of blood vessels in the brain and hindlimb was imaged at several time points (5 min, 3, 4, 6, 8, and 24 h) after the PCP-BDP2 NPs and ICG were injected via the tail vein, respectively. After 24 h, at the end of imaging, the mice were euthanized and their major organs containing the heart, liver, spleen, lung, kidney, and brain were isolated and imaged.

**Animals and tumor xenograft model.** We chose luciferase-expressed 4T1 tumors to establish the peritoneal carcinomatosis-bearing mouse model. One million cells were injected intraperitoneally for each mouse and the model could be formed in 5 days. All animal studies were performed according to the guidelines set by the Tianjin Committee of Use and Care of Laboratory Animals, and the overall project protocols were approved by the Animal Ethics Committee of Nankai University. The accreditation number of the laboratory is SYXK(Jin) 2019-0003 promulgated by Tianjin Science and Technology Commission.

**In vivo NIR-II fluorescence imaging of the lymphatic system.** Adult nude mice (6–8 weeks old) were selected and anesthetized with isoflurane and subsequently PCP-BDP2 NPs (50 μL, 1 mg/mL) was subcutaneous injected in the footpads. Images were recorded immediately at the time points of 5, 10, 30 min, 1–5, 7.5, and 24 h after injection of the NPs. Then the lymph nodes were dissected out and imaged by the guidance of NIR-II fluorescence imaging at 5 h, and then the excised lymph nodes were fixed in 4% paraformaldehyde, sectioned at a thickness of 6 μm, and stained with H&E. The slices were imaged by a digital microscope (Leica QWin).

**NIR-II fluorescence image-guided surgery.** The PCP-BDP2 NPs (200 μL, 1 mg/mL) were injected intravenously into the peritoneal carcinomatosis-bearing mice overnight. Before the surgery, D-luciferin suspended in PBS buffer was intraperitoneally injected into the mice first. About 5 min later, when the luciferase and luciferin are reacting completely, the mice were euthanatized and the abdomen cavity of mice was opened for subsequent Image-Guided Surgery. The Xenogen IVIS® Lumina II system was used for bioluminescence imaging, while the NIR-II fluorescence imaging was carried out on a home-built NIR-II imaging instrument. Then the removed tumor nodules were also stained with H&E for further analysis.

**Reporting summary.** Further information on research design is available in the Nature Research Reporting Summary linked to this article.

## Data availability

The data supporting the findings of this study are available from the authors on reasonable request, see author contributions for specific data sets. Source data are provided with this paper. The X-ray crystallographic coordinates for the structures reported in this article have been deposited at the Cambridge Crystallographic Data Centre (CCDC) under deposition numbers CCDC 2047734 and 2047735. These data can be obtained free of charge from The Cambridge Crystallographic Data Centre via www.ccdc.cam.ac.uk/data_request/cif.

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

## Acknowledgements

This work was financially supported by the National Natural Science Foundation of China (21971115, 51873092, 51961160730, and 21907050), the Natural Science Foundation of Jiangsu Province (BK20190282, BK20202004), the National Key R&D Program of China (Intergovernmental cooperation project, 2017YFE0132200), Tianjin Science Fund for Distinguished Young Scholars (19JCJQJC61200), the Science and Technology Project of Tianjin (20JCYBJC01140), the Fundamental Research Funds for the Central Universities, Nankai University. Z.L. and K.L. thank Dr. S. Gong and Dr. W. Duan for the helpful discussion on the design and synthesis of PCP-substituted BODIPY dyes.

## Author contributions

K.L., X.D., Z.J., D.D., Y.C., G.-Q.Z., and Z.L. conceived the experiments. Z.L., Y.C., and G.-Q.Z. prepared the paper. K.L., X.D., and Z.J. performed the experiments. All authors discussed the results and commented on the paper.

## Competing interests

The authors declare no competing interests.
