## [Peer Review File · Nature Communications]

REVIEWER COMMENTS

Reviewer #1 (Remarks to the Author):

The authors report the J-aggregation of an extended BODIPY dye (PCP-BDP2) applied for imaging in the NIR-II spectral region. The molecular design is well rationalized because the bulky substituent around the meso position avoids the stacking favoring the searched J-aggregation as demonstrated by x-ray packing. Besides, the photophysical characterization is exhaustive (extended and rich supporting information) and well explained with computational support. The formation of stable nanoparticles is a noteworthy point because they display NIR-I absorption and bright and stable NIR-II fluorescence (a long-lasting 6% efficiency at 1100 nm is really remarkable). Indeed, the biological essays in imaging (in vitro and in vivo) as well as in clinical surgery is notable and clearly demonstrated its improved performance in comparison with standard NIR-II probes.

Therefore, I think that the manuscript meets the required criteria of novelty and quality to deserve publication in Nature Communications. Moreover, I strongly recommend its publication because the conducted study (organic synthesis, photophysical properties, computational chemistry, in vivo and in vitro imaging essays) is really rigorous and appealing for the scientific community.

Just a minor remark about the ongoing ICT in PCP-BDP1. I agree that the PCP group could be able to induce ICT in meso position. However, the viscosity-dependent emission spectra of Figure S4 are rather striking in my opinion. In Figure S7 the relative intensity of the LE and ICT emission modestly change with the solvent polarity, but in Figure S4 the emission from the ICT drastically increases with the viscosity. The authors assign such trend to an inhibited rotation of PCP in viscous media. Such interpretation is logic. However, taking into account the size of the substituent and the methylation of the adjacent chromophoric positions, one could expect that the rotational motion of PCP is strongly hindered and has a low impact of the viscosity. Whereas the rest of the reported results are fully consistent from my point of view, these ones related to viscosity are not so clear.

Reviewer #2 (Remarks to the Author):

Liu and co-workers reported a meso-[2.2]Paracyclophanyl-BODIPY Dye (PCP-BDP2) with J-aggregation induced NIR II emission, which was applied for in vivo lymph node NIR II imaging and fluorescence-guided surgery. PCP-BDP2 shows a NIR I emission centered at 795 nm in dilute solution and NIR-II emission at 1010 nm in the J-aggregation state, while its control compound with phenyl in the meso-position of BODIPY (Ph-BDP2) exhibits NIR I around 750 nm in dilute solution and aggregation caused quenching. The introduction of PCP group with large steric hindrance effect not only endows stronger ICT process but also avoids the ACQ effect and allows the dye to form NIR II emitting J-aggregates. This work provided an effective strategy for developing imaging agents for NIR II imaging. The results are very interesting and I would like to recommend its publication after addressing the following minor issues:

1. In figure 1b, the black line noted as "s", which should better be changed to "M" as it stands for monomer emission in good solvent.
2. In Figure 2c, the authors measured the distance between BODIPY core and N,N-dimethylaminophenyl group as 3.3 Å, how about the distance between the two parallel BODIPY cores?

3. In line 149-150, the authors states that “the phenyl ring at the meso-position has a very weak conjugation effect on the BODIPY core because of its free rotation”, which should be change to “the phenyl ring at the meso-position has a very weak conjugation effect on the BODIPY core because of its perpendicular orientation toward BODIPY core”. In the same paragraph, the authors discussed about the rotation of meso-group to explain to different emission behavior of Ph-BDP1 and PCP-BDP1. However, the authors should pay more attention to the rotation abilities of the two molecules in their excited states rather than ground states, since the decay pathways of excited states are responsible for the emission behavior. In this regard, the author also need to discuss about the theoretical data such as changes on bond lengths, dihedral angles of some key atoms and rings.
4. In line 178-180, the authors states that “electronic transition of S1-S0 (ground state, $f = 0.0900$) can be ascribed to the CT emission, while the S2-S0 transition with relative large oscillator strength ($f = 0.4344$) is a fluorescent LE state”. The author should be careful before jump into that conclusion, since emission from higher excited states are usually hard to be observed according to Kasha’s rule. The LE-CT states transition could be realized by conformational change of the S1.
5. In Figure 2e, the FMOs in the right column should belongs to PCP-BDP2 not PCP-BDP1.
6. In line 208-212, the authors stated that higher solvent viscosity lead to the higher emission of PCP-BDP2, the authors should be careful since solvent polarity might also effect the emission intensity. To confirm the effect of viscosity, the viscosity dependent data of PCP-BDP2 should be provided like that of PCP-BDP1 in Figure S4.
7. Like Figure S9, The absorption spectra change of Ph-DBP1 and Ph-BDP2 in THF-water binary solvents varied fw should also be provide for understanding their ACQ behavior.
8. The absorption of solid film of PCP-BDP1 and PCP-BDP2 should better be provided for understanding the large red emission shift.
9. The biosafety data of the PCP-BDP2 nanoparticles should be provided since it is important for the evaluation of its potential for practical applications.
10. Since chiral PCP has R- and S- configuration, what about is the PCP-BDP1 and PCP-BDP2?
11. Compared with Ref 42,43 and 46, what’s the advantage of PCP-BDP1 and PCP-BDP2 as BODIPY J-aggregates? It will be better to summarize in a Table in SI.

Response to Reviewers' Comments

(Manuscript ID: NCOMMS-20-48465)

Reviewer #1:

The authors report the J-aggregation of an extended BODIPY dye (PCP-BDP2) applied for imaging in the NIR-II spectral region. The molecular design is well rationalized because the bulky substituent around the meso position avoids the stacking favoring the searched J-aggregation as demonstrated by x-ray packing. Besides, the photophysical characterization is exhaustive (extended and rich supporting information) and well explained with computational support. The formation of stable nanoparticles is a noteworthy point because they display NIR-I absorption and bright and stable NIR-II fluorescence (a long-lasting 6% efficiency at 1100 nm is really remarkable). Indeed, the biological essays in imaging (in vitro and in vivo) as well as in clinical surgery is notable and clearly demonstrated its improved performance in comparison with standard NIR-II probes.

Therefore, I think that the manuscript meets the required criteria of novelty and quality to deserve publication in Nature Communications. Moreover, I strongly recommend its publication because the conducted study (organic synthesis, photophysical properties, computational chemistry, in vivo and in vitro imaging essays) is really rigorous and appealing for the scientific community.

Response: Thanks a lot for the very positive comments! We have amended the manuscript according to the valuable comments.

I: Just a minor remark about the ongoing ICT in PCP-BDP1. I agree that the PCP group could be able to induce ICT in meso position. However, the viscosity-dependent emission spectra of Figure S4 are rather striking in my opinion. In Figure S7 the relative intensity of the LE and ICT emission modestly change with the solvent polarity, but in Figure S4 the emission from the ICT drastically increases with the viscosity. The authors assign such trend to an inhibited rotation of PCP in viscous media. Such interpretation is logic. However, taking into account the size of the substituent and the methylation of the adjacent chromophoric positions, one could expect that the rotational motion of PCP is strongly hindered and a low impact of the viscosity. Whereas the rest of the reported results are fully consistent from my point of view, these ones related to viscosity are not so clear.

We appreciate the helpful advice! After re-evaluation of the relationship between the rotation of PCP group and the photophysical properties, we agree with the reviewer's comments that rotation of PCP group should be strongly hindered. Therefore, our speculation that the viscosity-dependent fluorescence of PCP-BDP1 is ascribed to the inhibited rotation of PCP in viscous media is not appropriate. We corrected "the CT process was greatly enhanced due to the inhibited rotation of PCP-BDP1 in the high viscosity media" to "the CT process was greatly enhanced due to the inhibited intramolecular motion of PCP-BDP1 in the high viscosity media" in the revised manuscript.

We analyzed the difference of PCP-BDP1 geometry between the ground and photoexcited states by using the theoretical calculation results, and we think that the geometry rearrangement of PCP-BDP1 in the excited state should be responsible for the viscosity-dependent fluorescence of PCP-BDP1. In the ground state, the indacene plane of PCP-BDP1 is significantly bent to accommodate the van der Waals repulsion of the PCP group and the methyl groups at 1,7-positions. As a result, the boron atom as well as the carbon atom at PCP group, which are attached to the meso-position of the BODIPY core, deviated from the indacene plane with a distance around 0.26 Å and 0.49 Å, respectively. However, PCP-BDP1 shows distinct geometry rearrangement in comparison with that in the ground state. The indacene plane is further bent due to the photoexcitation. The deviation of the boron atom and the carbon atom at PCP group attached to the meso-position from the indacene plane increased to 0.38 Å and 0.84 Å,

respectively. This rearrangement in the excited state would ultimately favor the thermal relaxation of PCP-BDP1 in the first singlet state to an energetically hot, ground-state species. When PCP-BDP1 is in a highly viscous media, such geometry rearrangement would be efficiently inhibited. As a result, fluorescence intensity increment can be observed. These new results and related discussions have been added to the revised manuscript.

Reviewer #2:

Liu and co-workers reported a meso-[2.2]Paracyclophanyl-BODIPY Dye (PCP-BDP2) with J-aggregation induced NIR II emission, which was applied for in vivo lymph node NIR II imaging and fluorescence-guided surgery. PCP-BDP2 shows a NIR I emission centered at 795 nm in dilute solution and NIR-II emission at 1010 nm in the J-aggregation state, while its control compound with phenyl in the meso-position of BODIPY (Ph-BDP2) exhibits NIR I around 750 nm in dilute solution and aggregation caused quenching. The introduction of PCP group with large steric hindrance effect not only endows stronger ICT process but also avoids the ACQ effect and allows the dye to form NIR II emitting J-aggregates. This work provided an effective strategy for developing imaging agents for NIR II imaging. The results are very interesting and I would like to recommend its publication after addressing the following minor issues.

Response: Thanks a lot for the positive comments! We have amended the manuscript according to the valuable comments.

1. In figure 1b, the black line noted as “s”, which should better be change to “M” as it stands for monomer emission in good solvent.

We appreciate the advice. We have changed the letter “S” to “M” in revised Figure 1b.

2. In Figure 2c, the authors measured the distance between BODIPY core and N,N-dimethylaminophenyl group as 3.3 Å, how about the distance between the two parallel BODIPY cores?

The distance between the two parallel BODIPY cores was measured to be around 8.9 Å, which further confirmed the negligible interactions between the BODIPY cores.

3. In line 149-150, the authors states that “the phenyl ring at the meso-position has a very weak conjugation effect on the BODIPY core because of its free rotation”, which should be change to “the phenyl ring at the meso-position has a very weak conjugation effect on the BODIPY core because of its perpendicular orientation toward BODIPY core”. In the same paragraph, the authors discussed about the rotation of meso-group to explain to different emission behavior of Ph-BDP1 and PCP-BDP1. However, the authors should pay more attention to the rotation abilities of the two molecules in their excited states rather than ground states, since the decay pathways of excited states are responsible for the emission behavior. In this regard, the author also need to discuss about the theoretical data such as changes on bond lengths, dihedral angles of some key atoms and rings.

We appreciate the helpful advice. We have changed the sentence “the phenyl ring at the *meso*-position has a very weak conjugation effect on the BODIPY core because of its free rotation” to “the phenyl ring at the *meso*-position has a very weak conjugation effect on the BODIPY core because of its perpendicular orientation toward BODIPY core” in the amend manuscript.

We accepted your helpful suggestion and analyzed the optimized geometries in both ground and excited states to explore whether the photophysical properties difference is the result of geometry rearrangement in the excited state. As we discussed in the revised manuscript, Ph-BDP1 and Ph-BDP2 show almost the same geometries in both ground and excited states, which is similar to most of the classical BODIPY dyes. However, PCP-BDP1 and PCP-BDP2 show obvious geometry rearrangement in the excited state. This rearrangement would ultimately favor

the thermal relaxation of PCP-BDP1 and PCP-BDP2 in the first singlet state to an energetically hot, ground-state species. When PCP-BDP1 and PCP-BDP2 are in a highly viscous media, such geometry rearrangement can be efficiently inhibited. Therefore, the viscosity-dependent fluorescence intensity increment of PCP-BDP1 and PCP-BDP2 can be rationalized. These related discussions have been added to the revised manuscript.

4. In line 178-180, the authors states that “electronic transition of S1-S0 (ground state, $f = 0.0900$) can be ascribed to the CT emission, while the S2-S0 transition with relative large oscillator strength ($f = 0.4344$) is a fluorescent LE state”. The author should be careful before jump into that conclusion, since emission from higher excited states are usually hard to be observed according to Kasha’s rule. The LE-CT states transition could be realized by conformational change of the S1.

We appreciate the reminder! We agree with the comments that ascribe the S₂-S₀ transition to the LE state is not appropriate. We have deleted this ascription in the revised manuscript.

5. In Figure 2e, the FMOs in the right column should belongs to PCP-BDP2 not PCP-BDP1.

We appreciate the reminder and sorry for the mistake! We have corrected this mistake in the revised Figure 2.

6. In line 208-212, the authors stated that higher solvent viscosity lead to the higher emission of PCP-BDP2, the authors should be careful since solvent polarity might also effect the emission intensity. To confirm the effect of viscosity, the viscosity dependent data of PCP-BDP2 should be provided like that of PCP-BDP1 in Figure S4.

We appreciate the suggestion! The viscosity dependent spectra of PCP-BDP2 were measured. As shown in Figure S7 in the revised Supporting Information, PCP-BDP2 showed a 1.4-fold fluorescence intensity increase with the viscosity-increasing from 0.6 cp to 130 cP. This result further confirms that higher solvent viscosity can lead to the higher emission of PCP-BDP2.

7. Like Figure S9, the absorption spectra change of Ph-DBP1 and Ph-BDP2 in THF-water binary solvents varied f_w should also be provide for understanding their ACQ behavior.

We appreciate the advice and measured the absorption spectra of Ph-DBP1 and Ph-BDP2 in THF-water binary solvents with varied f_w . As shown in revised Figure S13, with the f_w increasing from 0 to 90%, the absorption bands of both Ph-BDP1 and Ph-BDP2 were redshifted and broadened, indicates the formation of H- and J-aggregates at the same time. Different from the aggregation caused J-aggregates emission enhancement of PCP-BDP1 and PCP-BDP2, the emission of both two compounds was gradually decreased, and no J-aggregates emission was observed. These results suggest that H-aggregation induced emission quenching dominated the photophysical properties of Ph-BDP1 and Ph-BDP2 in the aggregation state, which further demonstrated the key role of PCP group plays in the J-aggregation behavior of PCP-BDP1 and PCP-BDP2. The related discussions have been updated in the revised manuscript (Pages 11 and 12).

8. The absorption of solid film of PCP-BDP1 and PCP-BDP2 should better be provided for understanding the large red emission shift.

A8. We appreciate your advice! We prepared the thin film of PCP-BDP1 and PCP-BDP2 in quartz plate via standard spin-coating technique. As shown in Figure S12, PCP-BDP1 and PCP-BDP2 showed absorption

maximum at 532 and 774 nm, which are consistent with those observed in THF-water binary solvents at $f_w = 99\%$. This result further confirms the formation of J-aggregates in the aggregation state.

9. The biosafety data of the PCP-BDP2 nanoparticles should be provided since it is important for the evaluation of its potential for practical applications.

We appreciate your advice! We evaluated the cytotoxicity of PCP-BDP2 in L02 and HepG2 cells by standard MTT assay. As shown in Figure S15 in the revised Supporting Information, PCP-BDP2 with varied concentrations from 0.4 to 51.2 μM show negligible influence on the survival of L02 and HepG2 cells, suggesting its good biosafety.

10. Since chiral PCP has R- and S- configuration, what about is the PCP-BDP1 and PCP-BDP2?

The starting material PCP used to synthesize PCP-BDP1 and PCP-BDP2 is racemic. Future research on the relationship between the chiral structure of the PCP group and the photophysical properties/aggregation behavior of PCP-BDP1 and PCP-BDP2 may be very interesting.

11. Compared with Ref 42,43 and 46, what's the advantage of PCP-BDP1 and PCP-BDP2 as BODIPY J-aggregates? It will be better to summarize in a Table in SI.

We appreciate the advice! We summarized the photophysical properties of J-aggregates reported in references 42, 43 and 46 in Table S1 in the revised Supporting Information. In comparison with those BODIPY J-aggregates reported in references 42, 43 and 46, PCP-BDP1 and PCP-BDP2 show following advantages: 1) Tunable fluorescence from red to NIR-II. BODIPY J-aggregates reported in references 42, 43 and 46 show exhibited absorption/fluorescence in red or NIR region. The fluorescence of these J-aggregates cannot reach to the NIR-II region. 2) Aggregation induced J-aggregates emission enhancement. Aza-BODIPY and *meso*-carboxylate substituted BODIPY dyes show aggregation caused emission quenching, this feature makes these J-aggregates are not suitable for fluorescent imaging. In contrary, the introduction of PCP group can efficiently suppress the intermolecular interactions, resulting in the intense emission in the J-aggregation state.

REVIEWERS' COMMENTS

Reviewer #1 (Remarks to the Author):

I agree with the performed changes in the manuscript and the response to the issues addressed by the reviewers

Therefore I think that this updated version can be published without further action

Reviewer #2 (Remarks to the Author):

The authors have answered all the questions.

Response to Reviewers' Comments

(Manuscript ID: NCOMMS-20-48465A)

Reviewer #1:

I agree with the performed changes in the manuscript and the response to the issues addressed by the reviewers. Therefore I think that this updated version can be published without further action.

Response: Thanks a lot for the positive comments!

Reviewer #2:

The authors have answered all the questions.

Response: Thanks a lot for the positive comments!